# Research on Channel Characteristics and Electrode Electrical Performance of Earth Current Field Information Transmission Technology

**DOI:** 10.3390/s23135936

**Published:** 2023-06-26

**Authors:** Jingang He, Zhong Su, Zhan Xu, Zhe Kuang, Xiaowen Wen, Xin Zhou

**Affiliations:** 1School of Automation, Beijing Institute of Technology, Beijing 100081, China; 2China Communications Information Technology Group Co., Ltd., Beijing 101300, China; 3Beijing Key Laboratory of High Dynamic Navigation Technology, Beijing Information Science and Technology University, Beijing 100192, China; 4Beijing Galaxy Intelligent Digital Information Technology Co., Ltd., Beijing 102629, China

**Keywords:** grounding electrode, current field, information transmission, channel characteristics, electrical performance

## Abstract

Starting from the need for emergency rescue information transmission in tunnel engineering accidents, this article focuses on researching and solving the technical problems of information transmission between rescue personnel and trapped personnel after tunnel engineering collapse accidents, before and during the rescue process. The research objects are the information transmission channel and grounding electrode in the earth current field information transmission technology, and the electromagnetic characteristics of the earth medium and the electrical performance of the grounding electrode are studied and analyzed using the electromagnetic simulation software Maxwell based on finite element algorithm, establish a three-dimensional model based on the transmission of current field information of the ground electrode, analyze the effects of the electrode array, electrode depth, and radius on impedance. Research has shown that the impedance of the earth is related to the resistivity of the medium and is not a human-controllable factor. To reduce the contact impedance of an electric dipole antenna, one should start with the contact impedance of the earth electrode. The impedance of the transmitting end is an important factor affecting the efficiency of information transmission; parallel connection of multiple grounding electrodes, increasing the depth of grounding electrode penetration into the soil layer, and increasing the radius between grounding electrode pairs are all effective methods to reduce the contact impedance of electric dipole antennas, thereby improving information transmission capacity. To achieve wireless information transmission through the stratum, by appropriately selecting the operating frequency of electromagnetic waves, a certain distance of signal transmission can be achieved.

## 1. Introduction

Tunnel engineering has the characteristics of being “long, narrow, curved, and deep”, and belongs to limited space; During the tunnel excavation process, especially before the completion of the inverted arch and secondary lining, collapse accidents often occur; After the accident, the thickness of the collapsed rock and soil between the tunnel entrance and the construction face is generally 30–80 m. This type of “closed-door” collapse will form an underground enclosed space between the tunnel construction face and the collapsed body, causing the construction personnel on the tunnel face to be trapped. Because their external information transmission link is completely “cut off” by the collapsed body, it poses great difficulties for the rapid rescue of tunnel engineering. As shown in Figure 1.

Through comparative analysis of theoretical and practical research results in relevant fields both domestically and internationally, the use of earth electrode current field information transmission technology is currently the only effective way and method to solve the problem of information transmission between rescue personnel and trapped personnel during tunnel engineering collapse, pre rescue, and rescue processes.

The current field information transmission method of the earth electrode is a method of using extremely low-frequency electrical signals to be applied to two electrodes inserted into the earth’s soil layer, thereby forming a current field in the rock or soil layer. Wireless information transmission is achieved through signal detection at the receiving end. The essence of the earth electrode current field information transmission mode is to use the electric field current to carry information data. When the carrier frequency is low, the field current is mainly a conducted current, but when the carrier frequency is high, the field current is mainly a displacement current. At this time, the circuit formed by the wire, electrode, and earth is equivalent to a ring antenna, therefore the essence is also near-field electromagnetic wave information transmission.

The process of electromagnetic wave transmission in terrestrial channels is relatively complex, and it is influenced by non-uniform media. Electromagnetic waves interact with the media, producing various electromagnetic effects such as polarization, magnetization, and conduction. The impact of these electromagnetic effects on the transmission of electromagnetic waves is mainly evident in the reflection, refraction, and attenuation of electromagnetic wave signals. The inhomogeneity of the earth medium, the impact of geomorphic features, and multi-path transmission will cause signal distortion, attenuation, or changes in the direction of electromagnetic wave propagation. The strong penetration depth of the earth electrode current field information transmission method is related to factors such as the conductivity of the formation, signal frequency, current intensity injected into the earth, contact impedance between the electrode and the soil, and the material and shape of the electrode. The sensitivity of the receiving host is related to the distance between the receiving electrodes and the contact impedance between the electrode and the soil. The greater the distance between the two electrodes, the greater the detected voltage value; the smaller the contact impedance between the electrode and the soil, the higher the detected voltage value [1].

According to electromagnetic theory and transmission theory, in this article, we abstract the transmission process of the ground electrode current field information transmission signal into a transmission model, which can be composed of different parameters, such as soil dielectric constant, electrode depth, signal frequency, etc. A non-uniform earth medium can be described as a dielectric semiconductor. Due to the similarity between the characteristics of the electrostatic field in the semiconductive medium and the constant flow field in the conductive medium, when studying the propagation characteristics of the conductive current field in the conductive medium, it can be regarded as a quasi-electrostatic field at a certain moment. Therefore, when studying the channel characteristics and signal propagation laws based on the ground electrode current field, the relevant knowledge of the electrostatic field can be used for analysis [2].

## 2. Related Work

In recent years, many experts, and scholars from around the world have conducted research on the transmission technology of geomagnetic field information and its applications in related fields.

From 2009 to 2016, V Bataller and A Munoz et al. investigated the issue of contact impedance in a strong penetration information transmission mode based on current field propagation and pointed out that the effective information transmission distance is related to factors such as the conductivity of the rock and soil layer, the input signal frequency, the current intensity injected into the earth, and the material and shape of the electrode. In 2015, Van L. and Sunderman C., et al. studied a channel attenuation model based on electrode strong penetration information transmission. Considering noise interference and signal attenuation characteristics, the receiver sensitivity and transmitter power required for information transmission were calculated, and the field distribution model of the earth electrode strong penetration information transmission system in a uniform half-space was derived. The configuration scheme of the receiver and transmitter was provided. In 2016, Yan L, Zhou C, Miguel R et al. established and simulated a signal attenuation model to investigate the significant impact of signal attenuation characteristics on receiver sensitivity, transmitter antenna length and direction, and operating frequency on Earth. At the same time, a prototype was used for testing, and the results of the simulation and testing were very similar. In 2017, Yan L., Zhou C., Reyes M., et al. proposed a fully executable solution for three electric field components for electrode-based TTE information transmission. In 2017, Damiano N., Yan L., Whisner B., et al. studied the current injection and contact impedance characteristics of very low frequencies by analyzing recent simulation and measurement results from the National Institute of Occupational Safety and Health (NIOSH) in the United States and discussed the main factors affecting electrode based low-frequency information transmission systems. In 2017, Maxim Ralchenko, Mike Roper, et al. of the University of Carlton in Canada discovered the presence of low-frequency signals on two slender conductors near the transmitter, which may greatly increase the signal transmission distance. To better explore this phenomenon, they conducted many experiments and simulations. Simulate slender conductors with thin wires or use railway tracks or elevator shafts. By measuring the three magnetic field components near the conductor, it was found that there is a coupling effect in the signal, which is caused by the current inside the slender conductor. This discovery can be used to predict the feasibility of strong penetration information transmission links in different settings and optimize the positions of the transmitting and receiving ends [3,4,5,6,7].

In 2010, Wu Zhiqiang et al. designed an underwater electric field communication system based on a digital signal processor (DSP) and conducted experimental verification. In 2010, Wang Dandan et al. proposed using a constant flow field model as a model for underwater current field communication, obtained the relationship between input and output, and analyzed the advantages of current field communication methods. In 2012, Zhang Xinguo et al. analyzed and studied the effect of electrode size on the electric field of ring electrodes. In 2014, Su Baoping established an underwater current field communication channel model and analyzed the relationship between received signals, transmission voltage, communication distance, and other parameters. In 2015, Li Panfeng applied current field communication technology in the Xiaolangdi hydrological operation of the Yellow River, verifying the reliability of the technology. In 2016, Geng Weizhi et al. studied the conduction current field theory of electromagnetic information transmission based on the conduction current field theory and established a simulation model. Based on this, they completed the design and implementation of the transmitting end of the current field mine’s strong penetration information transmission system. It is believed that wireless information transmission of conducted current field is a relatively reliable and effective method for emergency information transmission. In 2021, Liu Baoheng conducted mathematical modeling and analysis of the current field through the ground communication electric field, achieving information transmission distances of over 300 m. In 2021, Zhang Yixin et al. proposed an equivalent circuit for communication channels based on the characteristics of water bodies. Through finite element simulation, the transmission path and electric field distribution of electrical signals were analyzed, and the effects of practical factors such as communication distance, electrode plate depth, water width, electrode plate size, water conductivity, and dielectric constant on the equivalent circuit parameters were simulated and calculated. In 2022, Liu Zhimin and others conducted a forward simulation of the focusing and deflection effects of DC-focusing multi-point power supply detection. According to the mechanism of focused electrical detection, the finite element method is used to solve the potential of each node in the spatial multipoint power field, and the electric field line differential equation of the spatial field is derived. In 2022, Wang Feng et al. conducted research on AUV path planning for optimal energy velocity in current fields. In 2023, Zhao Jun and others used finite element methods to solve the normal and abnormal potentials of spatial fields. Based on COMSOL Multiphysics software, a uniform three-dimensional geoelectric detection model was constructed and segmented using a tetrahedral mesh adaptive algorithm. We compared and analyzed the accuracy of numerical solutions, studied the distribution and changes of the focusing current field, and determined the range of influence of the focusing effect on the current ratio coefficient. In 2023, Xu Zhan et al. studied the transmission characteristics of wireless strong penetration information in the horizontal direction of the earth electrode current field based on an extremely low frequency horizontal dual electrode current field channel. Considering the effects of parameters such as electrode radius, electrode penetration depth, electrode spacing, transmission signal frequency, and signal transmission distance on the transmission performance of Through the Earth (TTE) communication, a TTE communication path loss model for the earth electrode current field was established, Analyzed the impact of various parameters on path loss and determined the optimal operating parameters for signal transmission. An extremely low-frequency current field TTE communication system was constructed based on selected parameters. The path loss of 3–10 Hz signals was tested at communication distances of 200 m and 400 m. The accuracy of the model was verified through comparative analysis of experimental data and simulation results [8,9,10,11,12,13,14,15,16,17,18].

This paper mainly conducts research on the strong penetration information transmission requirements in complex geological and electromagnetic environments in emergency rescue scenarios of tunnel engineering accidents. The research focuses on the channel characteristics and electrical performance of grounding electrodes in grounding current field information transmission technology, which is a key and difficult technology that needs to be broken through in grounding electrode current field information transmission technology. The research work of many experts and scholars from around the world provides a good idea and foundation for further technical research in this paper.

## 3. Characteristics of Current Field Information Transmission Channel for Earth Electrodes and Electromagnetic Wave Propagation Law

The current field information transmission channel of the grounding electrode studied in this article is the surrounding rock mass and collapsed medium of tunnel engineering in the earth. Compared with the traditional medium of information transmission technology, the attenuation, multipath, and interference characteristics of the earth medium on the signal are the difficulties in the research of current field information transmission of the grounding electrode. The transmission of current field information based on grounding electrodes is not only related to the parameters of the earth medium but also to the geometric characteristics of the underground space of the earth medium. Based on the transmission of current field information of grounding electrodes, low-frequency signals with strong penetrability are used, and layered earth media is used as the propagation medium. Electric waves pass through different rock layers from the emission point to the ground, and reflection and refraction occur after passing through each layer. The channel environment is very complex.

Assuming that the transmitting and receiving electrode pairs can be completely aligned (the field strength direction at the receiving point coincides with the baseline of the electric dipole), a channel model is constructed as shown in Figure 2, where $d_{tr}$ is the distance between the transmitting electrode, $l_{tr}$ is the depth of the transmitting electrode into the soil, $d_{re}$ is the distance between the receiving electrode, $l_{re}$ is the depth of the transmitting electrode into the soil, and $L_{e}$ is the penetration transmission distance.

### 3.1. The Mechanism of Earth Electrode Current Injection and the Electromagnetic Characteristics of Terrestrial Media

Earth electrode current injection refers to the injection of a certain frequency of current into the ground through one or more earth electrodes below the surface, resulting in a charged volume and the formation of an electric field within that volume. An electric field can cause the movement of underground charges, thereby causing the propagation of electromagnetic waves underground.

Earth electrode current injection has been widely used in underground communication, Geophysics, underground mineral exploration, and other fields.

The implementation of the earth electrode current injection mechanism is influenced by multiple factors, among which the most important factors include:(1)Electrode shape and material. During the process of current injection into the ground electrode, the shape of the electrode and the quality of the material determine the effect of current injection into the ground and the intensity of the generated electric field.(2)The frequency and intensity of the current. The frequency and intensity of the current have a significant impact on the movement of charges injected into the ground. The frequency and intensity of the current should not be too high, as excessive current can cause serious electromagnetic interference.(3)The medium and terrain of the earth’s rock formations. The medium of geological strata and terrain has a significant impact on the propagation rate and amplitude of electromagnetic waves. Some special underground rock formations or terrain conditions can produce phenomena such as reflection and refraction, which affect the propagation rate and amplitude of electromagnetic waves.

In this article, To simplify the research and analysis process, the earth is assumed to be an infinitely uniform conductive medium, we use three physical quantities to describe the electromagnetic characteristics of the earth medium: magnetic permeability $\mu \left(H/m\right)$, dielectric constant $\varepsilon \left(F/m\right)$, and conductivity $\sigma \left(s/m\right)$.

(1)Magnetic permeability $\mu \left(H/m\right)$.Magnetic permeability is used to characterize the magnetic characteristics of a medium, as both soil and rock layers are non-magnetic media. Therefore, the earth’s magnetic permeability is set to be the same as in the air, $\mu =4\pi \times 10^{-7}\;H/m$.(2)Dielectric constant $\varepsilon \left(F/m\right)$. The value of the earth’s dielectric constant is related to rock composition, water content, salt content, and temperature [19]. (3)Conductivity $\sigma \left(s/m\right)$.In the underground communication channel of the earth, conductivity will cause energy attenuation to the electromagnetic waves transmitted in the equation. The influencing factors of conductivity in rock layers include water content, temperature, and mineral composition. In addition, frequency has a certain impact on conductivity, but when the frequency changes between 5 Hz and 5000 Hz, the conductivity hardly changes with frequency [20,21,22]. The resistivity ρ is the reciprocal of the conductivity σ.

### 3.2. Propagation of Electromagnetic Waves in Earth Medium in the Current Field of Grounding Electrodes

Electromagnetism theory points out that both the electrostatic field and steady current field can be described by two groups of corresponding physical quantities as shown in Table 1. On the premise of meeting certain conditions, both groups of physical quantities follow the physical laws with the same mathematical form.

In the passive region, the electrostatic field in an isotropic medium satisfies differential equations and boundary conditions:(1)$$\begin{array}{l} \nabla •D=0\;\;D_{1n}=D_{2n}\\ \nabla \times E=0\;\;E_{1t}=E_{2t} \end{array}$$*n* represents the normal direction, and *t* represents the tangential direction. Subscripts 1 and 2 represent the media on both sides of the boundary. If a potential function is used *φ* to describe the electric field, its field equation is $\nabla^{2}\varphi =0$ (Laplace’s equation, space free), and the corresponding boundary conditions are:(2)$$\varphi_{1}=\varphi_{2},\varepsilon_{2}\frac{\partial \varphi_{2}}{\partial n}=\varepsilon_{1}\frac{\partial \varphi_{2}}{\partial n}$$

Similarly, the stable current field in an isotropic conductive medium satisfies differential equations and boundary conditions:(3)$$\begin{array}{l} \nabla •j=0,\;j_{1n}=j_{2n}\\ \nabla \times E=0,\;E_{1t}=E_{2t} \end{array}$$

If the potential function *j* is used to describe the electric field, the field equation is $\nabla^{2}\varphi =0$ (Laplace’s equation), and the corresponding boundary conditions are:(4)$$\varphi_{1}=\varphi_{2},\;\sigma_{2}\frac{\partial \varphi_{2}}{\partial n}=\sigma_{1}\frac{\partial \varphi_{2}}{\partial n}$$

Comparing the physical quantities of the above two groups, it can be found that *D*, *E*, *ε* and *j*, *E*, *σ* one by one correspondence, have the same form of vector field equations and boundary conditions. Whether it is a constant current field or an electrostatic field, the distribution of the potential in the medium can be expressed by Laplace’s equation.

For electrodes made of good conductors, their surface potential is uniformly distributed, so the two fields meet the same type of boundary conditions. If $\varepsilon_{2}/\varepsilon_{1}=\sigma_{2}/\sigma_{1}$, the electrostatic field distributed in a certain conductive medium can be used to simulate the current field with the corresponding dielectric.

If a plane electromagnetic wave with a time-dependent form of $e^{j\omega t}$ propagates in an infinite, uniform semi-conductive medium, Maxwell’s equation in its complex form is:(5)$$\nabla \times \bar{H}=-j\omega \varepsilon_{r}\bar{E}$$
(6)$$\nabla \times \bar{E}=-j\omega \mu \bar{H}$$

$\varepsilon_{k}=\varepsilon -j\frac{\sigma }{\omega }$ is the complex permittivity. The vector wave equation can be obtained by rotating the above formula. The propagation of electromagnetic waves in the tunnel collapse medium can be expressed by the following formula:(7)$$\bar{E}=E_{0}e^{-jkz}=E_{0}e^{-\beta z}e^{-j\left(\omega t+\alpha z\right)}$$
(8)$$\bar{H}=H_{0}e^{-jkz}=H_{0}e^{-\beta z}e^{-j\left(\omega t+\alpha z\right)}$$

In Formulas (7) and (8):

$E_{0}$ is a constant vector, representing the initial amplitude and polarization direction of the electric field vector;

Z is the vector in the propagation direction;
(9)$$k=\sqrt{j\omega \mu \left(\sigma +j\omega \varepsilon \right)}=j\omega \sqrt{\mu \left(\varepsilon -j\sigma /\omega \right)}=\alpha +j\beta$$

In Formula (9):

α is the phase constant, $\alpha =\omega {\left\{\frac{\mu \varepsilon }{2}\left[1+\sqrt{1+{\left(\sigma /\omega \varepsilon \right)}^{2}}\right]\right\}}^{\frac{1}{2}}$;

β is called the attenuation constant, $\beta =\omega {\left\{\frac{\mu \varepsilon }{2}\left[-1+\sqrt{1+{\left(\sigma /\omega \varepsilon \right)}^{2}}\right]\right\}}^{\frac{1}{2}}$;

When $\left|E/E_{0}\right|=1/e$, obtain the penetration transmission distance:(10)$$L_{e}=1/\beta =\sqrt{2}/{\left\{\omega \mu \varepsilon \left[-1+\sqrt{1+{\left(\sigma /\omega \varepsilon \right)}^{2}}\right]\right\}}^{\frac{1}{2}}$$

When the medium is conductive, due to σ/ωε>>1, ω=2πf, the penetration transmission distance can be expressed as
(11)$$L_{e}=1/\sqrt{\pi f\mu \sigma }=503\sqrt{\frac{\rho }{f}}$$

In Formula (11)

μ is the magnetic permeability, $\mu =4\pi \times 10^{-7}\;H/m$; 

f is the frequency, Hz;

ρ is the formation resistivity, ρ=1/σ, Ω·m.

Select some typical resistivity values (ρ=20 Ω·m, ρ=60 Ω·m, ρ=100 Ω·m, ρ=150 Ω·m), and simulate the relationship between the corresponding penetration transmission depth, resistivity, and operating frequency as shown in Figure 3.

This indicates that the penetration ability of electromagnetic waves is related to the frequency f and the resistivity ρ of the medium. The lower the frequency f, the greater the resistivity ρ of the medium, and the farther the propagation distance. However, the higher the resistivity p of the medium, the more severe the attenuation of electromagnetic waves by the formation, and the weak signal at the receiving point. Therefore, in order to achieve wireless information transmission through the stratum, by appropriately selecting the operating frequency of electromagnetic waves, a certain distance of signal transmission can be achieved.

## 4. Impedance Characteristics of Grounding Electrode

In a current field information transmission system based on grounding electrodes, the impedance of the transmitting end mainly consists of three parts: wire impedance, earth impedance, and grounding electrode impedance, as shown in Figure 4.

Wire impedance refers to the impedance generated by the wire connected between the current field-generating device and the grounding electrode. Usually, the wire length is short and high-performance transmission lines can be used to optimize the wire transmission effect. Therefore, wire impedance can be ignored in the transmitter impedance. Earth impedance refers to the impedance formed by the earth medium between the grounding electrodes at the transmitting end. The value of the earth impedance is relatively small and is often ignored. Therefore, the fundamental factor determining the impedance of the transmitting end is the impedance of the grounding electrode formed by the contact between the grounding electrode and the earth medium.

The impedance of the transmitting end is an important factor affecting the efficiency of information transmission. In the information transmission system of the earth electrode current field, the impedance of the transmitting end can be considered as a pure resistance.

According to the path loss formula $PL=-20\mathrm{lg}\frac{U_{re}}{U_{tr}}$, the path loss of the transmitting and receiving ends can be expressed as:(12)$$PL\left(d_{tr},d_{re},Z,L_{e},f,\sigma \right)=-20\mathrm{lg}\left(\left|\frac{1}{Z}\cdot \frac{d_{tr}d_{re}}{4\pi \sigma {L_{e}}^{3}}\cdot e^{-\beta L_{e}}\right|\right)$$

Therefore, according to Formula (12), assuming $L_{e}=100\;m$, transmission frequency f=10 Hz, and electrode pair spacing $d_{tr}=d_{re}=4\;m$, the relationship between the impedance of the transmitting end and the path loss can be obtained, as shown in Figure 5.

From Figure 5, it can be seen that the path loss significantly increases with the increase in the transmitter impedance. Therefore, the transmitter impedance should be reduced as much as possible to optimize the transmission effect.

### 4.1. Influence of Electrode Array on Impedance

According to Figure 2 schematic diagram of channel model and Figure 4 impedance composition at the transmitting end, the contact impedance $Z_{e}$ of a pair of electrodes has the following model:(13)$$Z_{e}=\frac{1}{4\pi \sigma l}\ln\left(\frac{4l}{r}-1+\frac{1}{d}\right)$$

In Formula (13):

σ is the conductivity of the surrounding rock mass and collapse medium of the tunnel, Ω·m;

l is the burial depth of the grounding electrode, m; 

r is the radius of the grounding electrode, m;

d A is the separation distance between two parallel electrode pairs, m.

This model reflects the effects of the depth of grounding electrode insertion into the soil layer, electrode radius, electrode pair spacing, and dielectric conductivity on the impedance of the grounding electrode.

When multiple sets of grounding electrode pairs are used to form an array grounding electrode, the array electrode arrangement is shown in Figure 6.

This connection method is equivalent to the parallel connection of multiple sets of grounding electrode pairs, and its circuit model is shown in Figure 7.

The antenna array is formed by increasing the number of grounding electrodes. The working principle of the antenna array can be seen as the superposition of electromagnetic waves (electromagnetic fields).

If an array electrode is composed of *n* sets of grounding electrode pairs, the impedance of the array electrode $Z_{en}$ can be expressed as:(14)$$Z_{en}=\frac{1}{n/Z_{e}}=\frac{Z_{e}}{n}=\frac{1}{4n\pi \sigma l}\ln\left(\frac{4l}{r}-1+\frac{1}{d}\right)$$

Taking the conductivity σ=0.01 s/m of the surrounding rock mass and collapsed body of the tunnel, according to Formula (14), assuming that the depth of the grounding electrode into the soil is *l* = 1.5 m, the spacing between the grounding electrode pairs is *d* = 4 m, and the radius of the grounding electrode is *r* = 1 cm, the relationship between the number of grounding electrode pairs and the impedance of the array electrode is simulated as shown in Figure 8.

From Figure 8, it can be seen that the impedance of the array electrode decreases as the number of grounded electrode pairs increases.

### 4.2. Influence of Electrode Depth and Electrode Pair Spacing on Impedance

Taking the conductivity σ=0.01 s/m of the surrounding rock mass and landslide mass of the tunnel, according to Formula (13), the spacing between grounding electrode pairs is set to *d* = 4 m, and the electrode radius *r* = 1 cm. The relationship between the depth of grounding electrode penetration and the impedance of grounding electrode pairs is simulated as shown in Figure 9.

Taking the conductivity σ=0.01 s/m of the surrounding rock mass and landslide mass of the tunnel, according to Formula (13), set the depth of the electrode pair into the soil as *l* = 1.5 m, and the spacing between the grounding electrode pairs as *d* = 4 m. The relationship between the radius of the grounding electrode pair and the impedance of the grounding electrode pair is simulated as shown in Figure 10.

Figure 10b shows the local interval data of (a).

Taking the conductivity σ=0.01 s/m of the surrounding rock mass and landslide mass of the tunnel, according to Formula (13), set the depth of the electrode pair into the soil as *l* = 1.5 m, and the electrode radius *r* = 1 cm. The relationship between the distance between grounding electrode pairs and the impedance of grounding electrode pairs is simulated as shown in Figure 11.

From Figure 9 and Figure 10, it can be seen that the impedance of the grounding electrode decreases significantly with the increase of the depth and radius of the grounding electrode; Increasing the contact area between the grounding electrode and the surrounding rock mass and collapse medium of the tunnel can effectively reduce the impedance of the transmitting end and improve the transmitting effect.

Meanwhile, as shown in Figure 11, the impedance change caused by increasing the spacing between grounding electrode pairs is very small, indicating that increasing the spacing between electrode pairs is not an effective way to reduce impedance.

## 5. Simulation and Testing Analysis

In this study, we established a three-dimensional electric field model for the transmission of current field information at the ground electrode for different signal enhancement methods. Taking the electrodes in the ground current field information transmission technology as the research object, a three-dimensional model based on the transmission of the ground electrode current field information was established using the electromagnetic simulation software ANSYS MAXWELL based on the finite element algorithm. The influence of the buried depth radius and electrode array layout on contact impedance was analyzed.

We performed 3D modeling according to the parameters in Table 2.

As shown in Figure 12, we imported the 3D geometric model of current field strength penetration information transmission into ANSYS MAXWELL.

The use of finite element theory and methods to calculate the electromagnetic distribution of a strong penetrating current field involves the setting and discussion of physical model boundary conditions and excitation. As mentioned earlier, when studying the propagation characteristics of the current field, it can be regarded as a constant flow field. Therefore, when setting the excitation, a constant voltage value of ±100 V can be applied to the two electrodes at the emission end, as shown in Figure 13a. Further, in order to simulate the real electromagnetic environment, it is necessary to introduce virtual boundaries to truncate the infinite area to a fixed space. Therefore, a vacuum area was set outside the entire collapse body, with the surface set as insulation, as shown in Figure 13b.

To ensure calculation accuracy, the number of iterations was set to 15, and each time the grid was automatically encrypted by 40%. The results of grid division are shown in Figure 14.

The distribution of the electric field in the entire earth’s medium was obtained from the simulation results, as shown in Figure 15. To analyze the impedance of the transmitting end, a cylindrical cross-section with a height of 2 m and a radius of 1 m was set up, and the positive electrode of the transmitting end was completely wrapped around the grounding electrode, as shown in Figure 16. Since all the current at the transmitting end is emitted from the positive electrode grounding, it can be considered that the current passing through this cross-section is the total current at the transmitting end.

Due to the different propagation directions of current in the earth’s medium, it is necessary to calculate the current density through a cylindrical cross-section. The relationship between current density and current value is:(15)$$J=\frac{I}{S}$$

In Formula (15):

*J* is the current density;

*I* is the current value;

*S* is the cross-sectional area of the conductor.

The entire cross-section through which the current needs to be calculated is a cylindrical surface, as shown in Figure 14b. Therefore, in the Maxwell field calculator, it is necessary to follow the integration formula
(16)$$I=\iint \vec{J}\cdot d\vec{S}$$

Let us integrate the current density of the entire cylinder in the direction of its normal vector to obtain the current value. Due to the known voltage of the two electrodes at the emission end and the current emitted by the electrodes, according to Ohm’s law:(17)$$R=\frac{U}{I}$$

In Formula (17):

*R* is the conductor resistance value;

*U* is the voltage at both ends of the conductor;

*I* is the current in the conductor.

Substituting the excitation voltage and the calculated current value can make it possible to obtain the resistance value between the two electrodes at the emission end.

The length of electrode pairs entering the soil layer and the impedance of the emitter electrode.

Table 3 records when the electrical resistivity σ=0.01 s/m, the radius of the ground electrode r=1 cm, the number of ground electrode pairs n=1, the relationship between the length of electrode pairs entering the soil layer, and the impedance of the emitter electrode.

Table 4 records when the electrical resistivity σ=0.01 s/m, the length of electrode pairs entering the soil layer l=1.2 m, the number of ground electrode pairs n=1, and the relationship between the radius of the earth electrode and the impedance of the earth electrode.

Table 5 records when the electrical resistivity σ=0.01 s/m, the length of electrode pairs entering the soil layer l=1.2 m, the radius of the ground electrode r=1 cm, and the relationship between the number of array pairs of ground electrodes and the impedance of the ground electrode.

From the records in Table 3, Table 4 and Table 5, it can be seen that the theoretical simulation results are not significantly different from the Maxwell finite element simulation results in terms of numerical values and the trend is consistent. That is, the greater the depth and radius of the grounding electrode, the more pairs of electrodes, and the smaller the impedance of the transmitting end. 

The change in impedance $Z_{e}$ of the ground electrode is ultimately reflected in the change in the received voltage value $U_{re}$ detected by the receiving end. Combined with Formula (14), the relationship is:(18)$$U_{re}=\frac{U_{tr}}{Z_{e}}\cdot \frac{d^{2}}{4\pi \sigma {L_{e}}^{3}}\cdot e^{-\beta L_{e}}=\frac{U_{tr}nld^{2}e^{-\beta L_{e}}}{{L_{e}}^{3}\cdot \ln\left(\frac{4l}{r}-1+\frac{1}{d}\right)}$$

In Formula (18),

$\beta =\omega {\left\{\frac{\mu \varepsilon }{2}\left[-1+\sqrt{1+{\left(\sigma /\omega \varepsilon \right)}^{2}}\right]\right\}}^{\frac{1}{2}}=2\pi f{\left\{\frac{\mu \varepsilon }{2}\left[-1+\sqrt{1+{\left(\sigma /2\pi f\varepsilon \right)}^{2}}\right]\right\}}^{\frac{1}{2}}$;

$U_{tr}$ is the emission voltage, V;

σ is the conductivity of the surrounding rock mass and collapse medium of the tunnel, S/m;

$L_{e}$ is the penetration transmission distance, m;

d is the distance between electrode pairs, m;

In order to verify the correctness of the impedance model of the earth electrode studied in this article, on-site testing was conducted in a tunnel under construction in Beijing. The distance between the transmitting and receiving ends of the test is $\;L_{e}=172\;m$, the distance between the grounding electrode d=4 m, the transmission frequency f=10 Hz, and the transmission voltage $U_{tr}=200\;V$. 

The testing structure is shown in Figure 17, and the testing site is shown in Figure 18.

Set the radius of the ground electrode to r=1 cm, change the depth of the ground electrode pair, measure and record the voltage value received by the receiving end; Set the depth of the ground electrode into the soil layer to l=1.2 m, change the radius of the ground electrode, measure and record the voltage value received by the receiving end; Compare the measured values with the theoretical values calculated according to Formulas (13) and (18), as shown in the Figure 19 and Figure 20.

From the comparative data, it can be seen that there is some deviation between the actual voltage detected at the receiving end and the theoretical calculation value. The reason for this is that existing geological exploration methods cannot accurately describe the real geological situation. Therefore, there must be a certain deviation between the equivalent uniform volume resistivity value calculated based on existing geological exploration data and the actual earth medium resistivity value, which also reflects the complexity of electromagnetic wave propagation in the earth medium.

The grounding electrode impedance Formula (13) and array electrode impedance Formula (14) proposed in this paper were verified through ANSYS MAXWELL simulation and practical application testing in engineering sites. By increasing the depth and radius of the ground electrode pair entering the soil layer, or using an electrode pair array, the impedance of the ground electrode can be effectively reduced, and the voltage value received by the receiving end can be significantly enhanced, thereby improving system performance.

## 6. Conclusions

This research was based on the information transmission technology of earth electrode current field. We studied and analyzed the electromagnetic characteristics of the earth medium, the shape of the grounding electrode, the influence of the electrode array on impedance, and the influence of electrode depth and radius on impedance. Using the electromagnetic simulation software Maxwell, based on the finite element algorithm, a three-dimensional model based on the transmission of current field information of the ground electrode was established to analyze the influence of the depth radius and electrode array arrangement on contact impedance.

Our research has shown the following:(1)The impedance of the earth is related to the resistivity of the medium, and it is not a human-controllable factor. To reduce the contact impedance of the electric dipole antenna, the contact impedance of the earth electrode should be considered.(2)The impedance of the transmitting end is an important factor affecting the efficiency of information transmission.(3)By increasing the depth and radius of the ground electrode pair entering the soil layer, or using an electrode pair array, the impedance of the ground electrode can be effectively reduced, and the voltage value received by the receiving end can be significantly enhanced, thereby improving system performance.(4)To achieve wireless information transmission through the stratum, by appropriately selecting the operating frequency of electromagnetic waves, a certain distance of signal transmission can be achieved.

Based on the research results, we will apply them to the development of an emergency rescue information transmission system for the earth electrode current field; When the transmission frequency is set to 10 Hz, the applied electrode voltage is DC250V, and the electrode is inserted into the formation at a depth of 1.5 m, we have achieved a penetration of 1000 m of the earth’s formation and transmitted text information of no more than 24 characters. The earth electrode current field information transmission technology is an emerging information transmission technology. In addition to being mainly applied in the field of emergency rescue communication in underground engineering, the related technology has broad application prospects in geological exploration, groundwater detection, soil moisture content detection, disaster monitoring, and other fields. With the development of information technology and the increasing demand for practicality, its application scope and prospects are also becoming increasingly widespread. However, in terms of the scope covered in this article, the work carried out is still limited, and further in-depth research on technical issues is needed in the future based on continuous exploration and practice. Based on existing research results and future application needs, we have reason to believe that the earth electrode electric field information transmission technology will be widely applied and promoted in future research and practice and contribute to emergency rescue services worldwide.

## Figures and Tables

**Figure 1 sensors-23-05936-f001:**
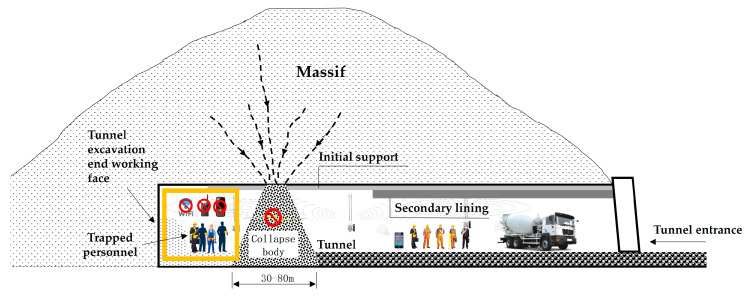
Schematic diagram of collapse accident status in tunnel engineering.

**Figure 2 sensors-23-05936-f002:**
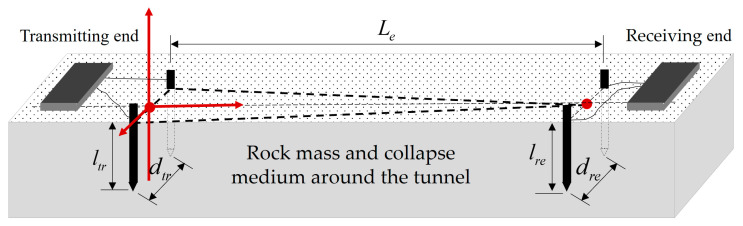
Schematic diagram of channel model and transceiver electrode pairs.

**Figure 3 sensors-23-05936-f003:**
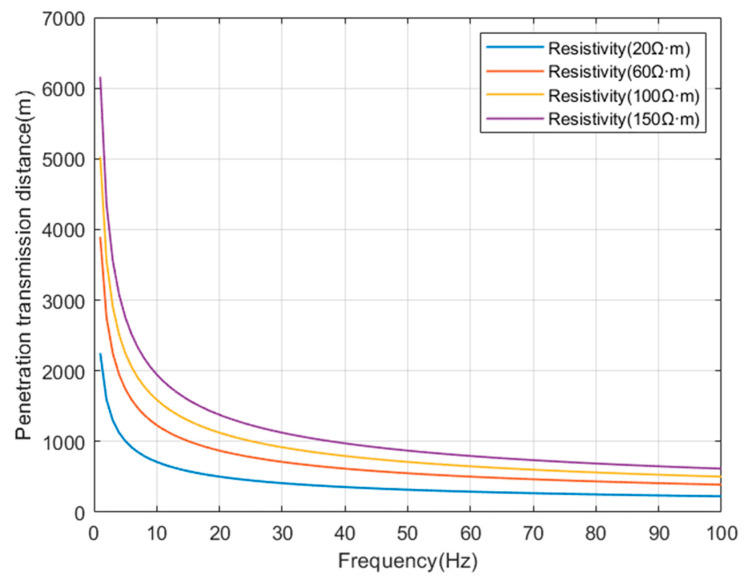
The relationship between penetration transmission depth, resistivity, and operating frequency.

**Figure 4 sensors-23-05936-f004:**
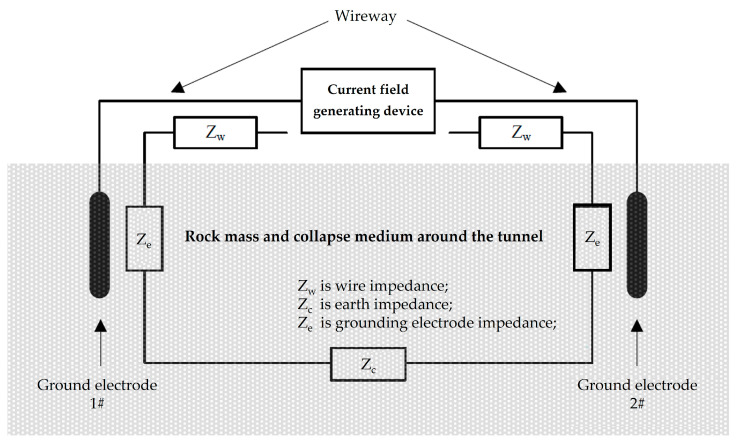
Impedance composition at the transmitting end.

**Figure 5 sensors-23-05936-f005:**
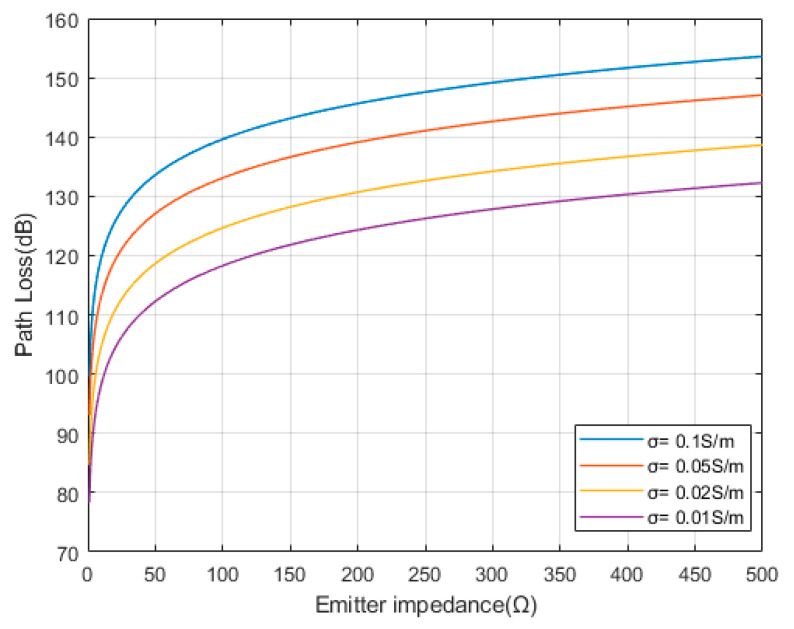
Relationship between emitter impedance and path loss.

**Figure 6 sensors-23-05936-f006:**
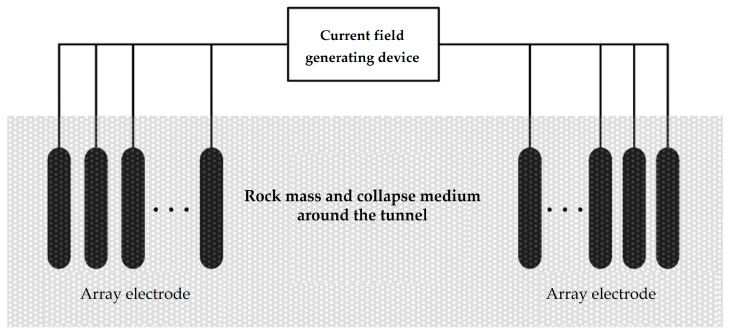
Array electrode layout.

**Figure 7 sensors-23-05936-f007:**
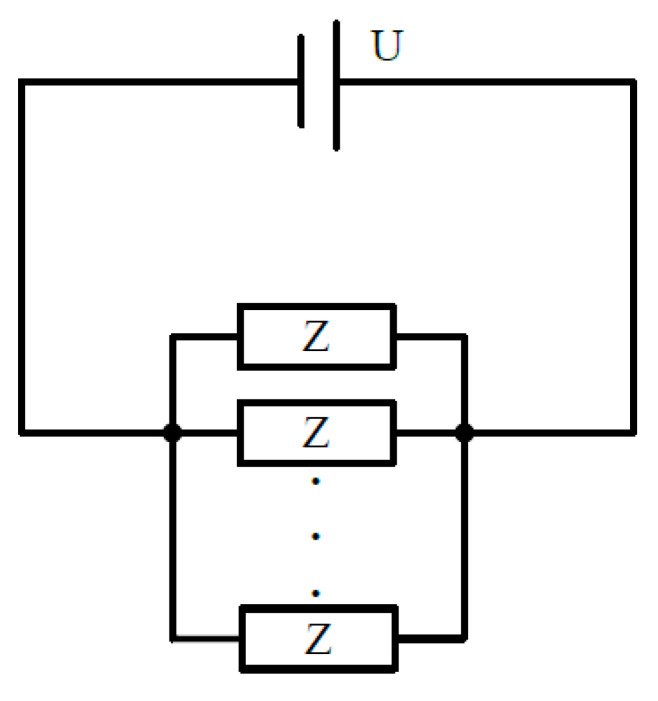
Array electrode circuit model.

**Figure 8 sensors-23-05936-f008:**
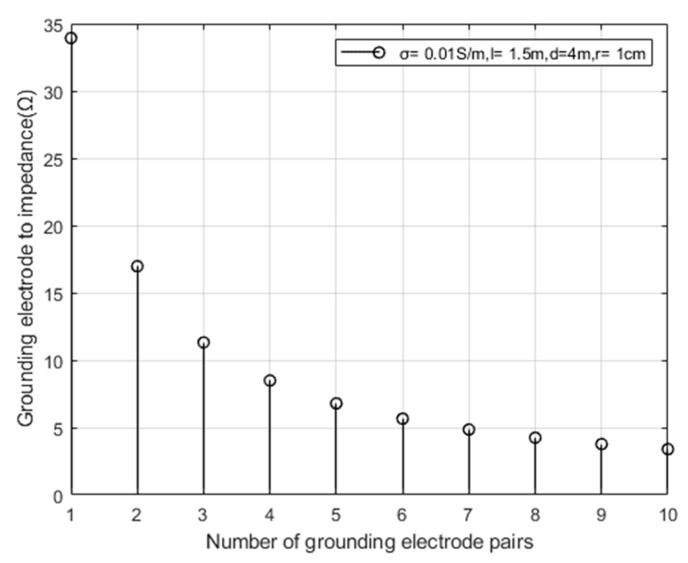
Relationship between the number of grounding electrode pairs and the impedance of array electrodes.

**Figure 9 sensors-23-05936-f009:**
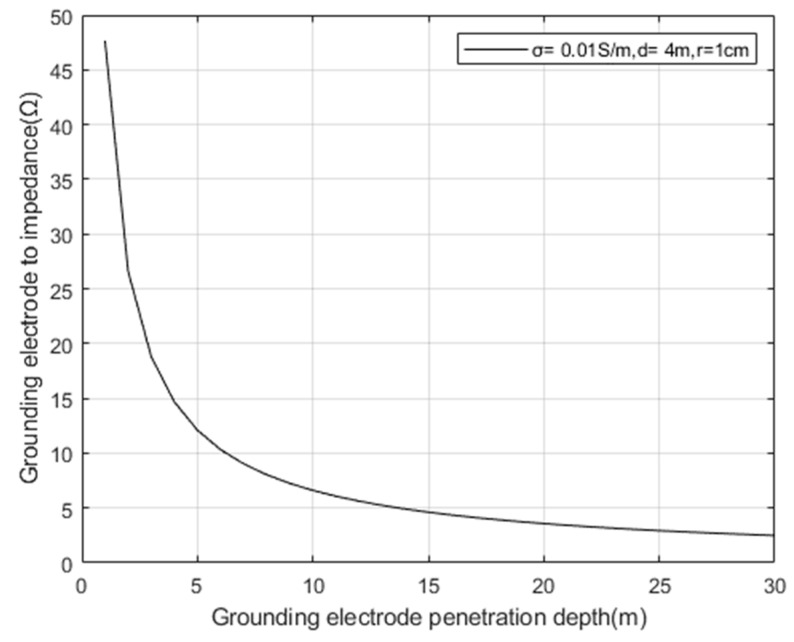
Relationship between grounding electrode penetration depth and grounding electrode impedance.

**Figure 10 sensors-23-05936-f010:**
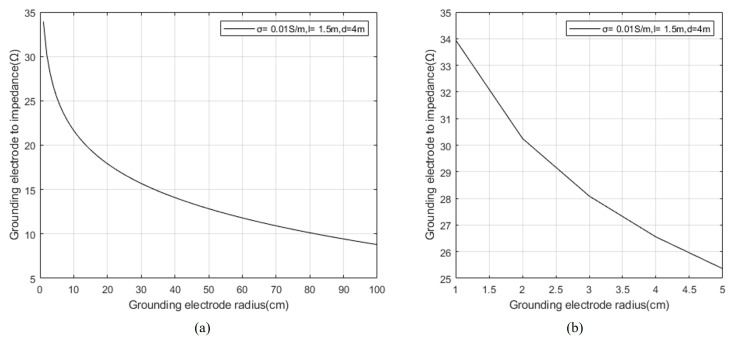
Relationship between the radius of the grounding electrode and the impedance of the grounding electrode.

**Figure 11 sensors-23-05936-f011:**
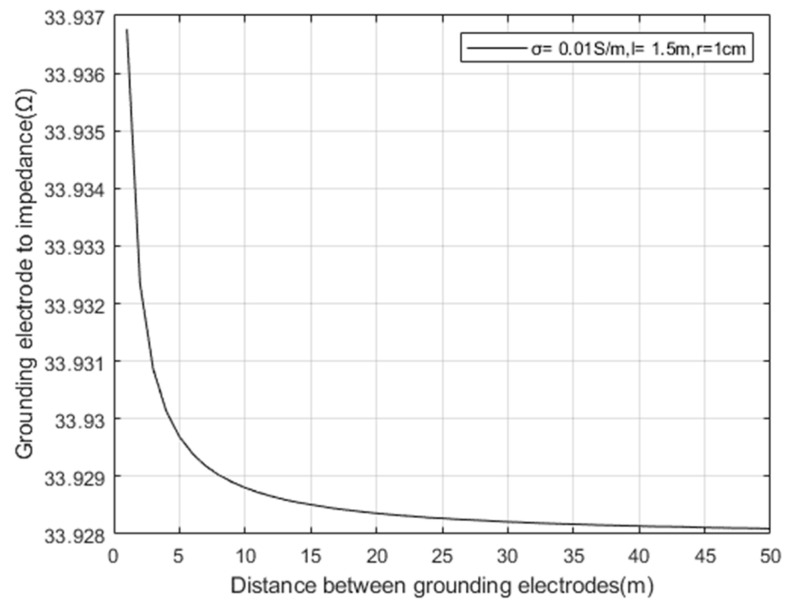
Relationship between the spacing of grounding electrode pairs and the impedance of grounding electrode pairs.

**Figure 12 sensors-23-05936-f012:**
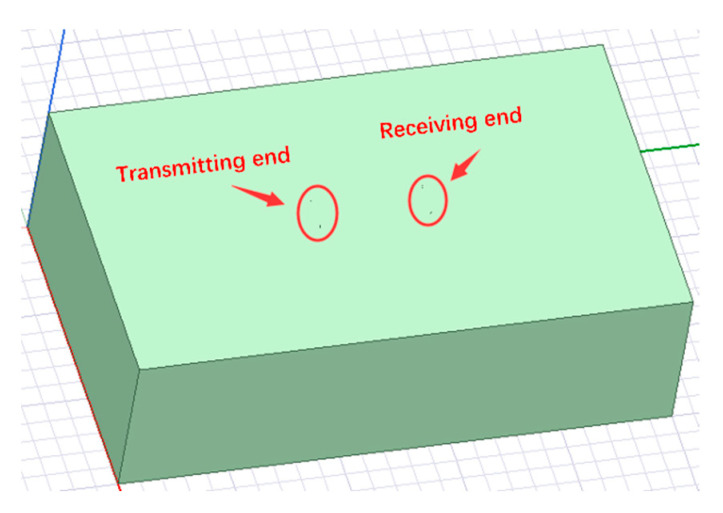
Simulation model for information transmission of earth electrode current field in Maxwell.

**Figure 13 sensors-23-05936-f013:**
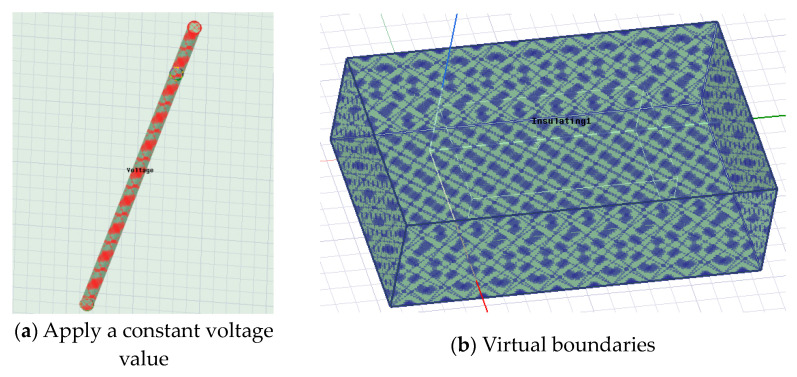
Excitation and boundary conditions.

**Figure 14 sensors-23-05936-f014:**
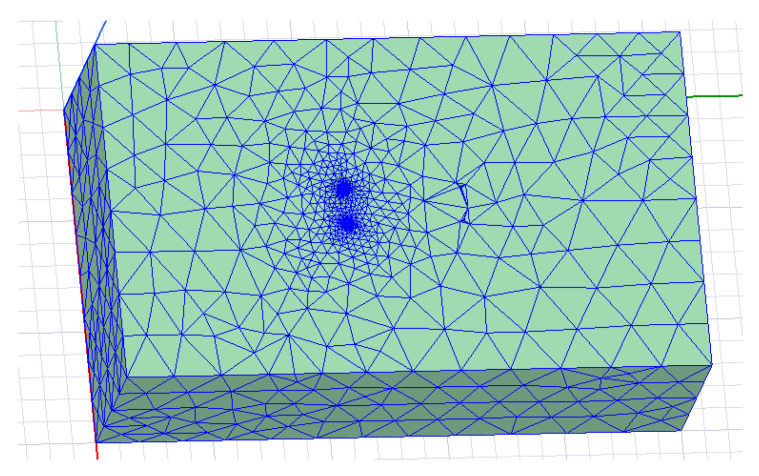
Adaptive mesh generation.

**Figure 15 sensors-23-05936-f015:**
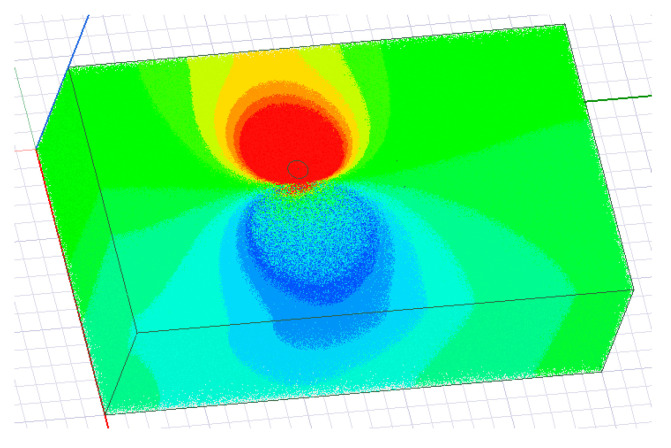
Distribution of electric field in the earth’s medium.

**Figure 16 sensors-23-05936-f016:**
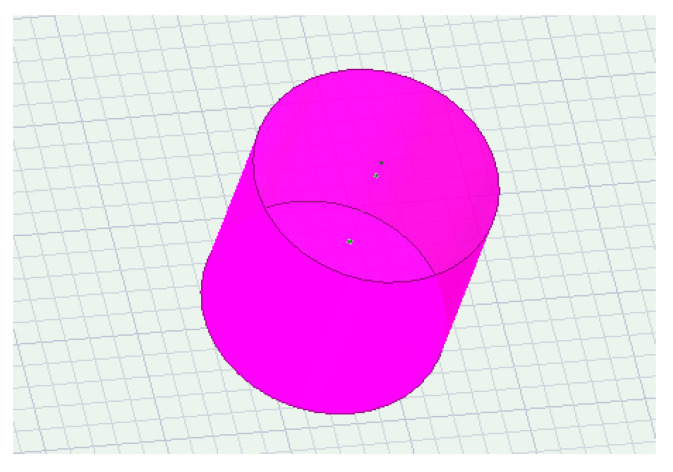
Wrapped emitter electrode.

**Figure 17 sensors-23-05936-f017:**
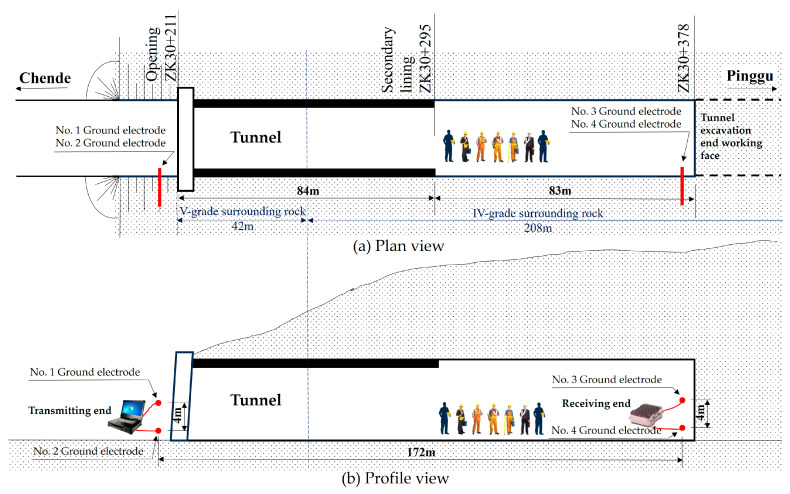
Testing Structure.

**Figure 18 sensors-23-05936-f018:**
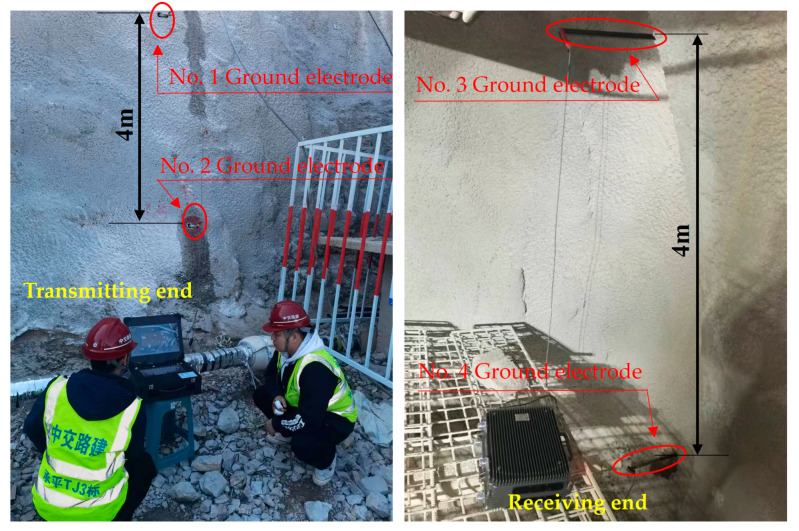
Testing site.

**Figure 19 sensors-23-05936-f019:**
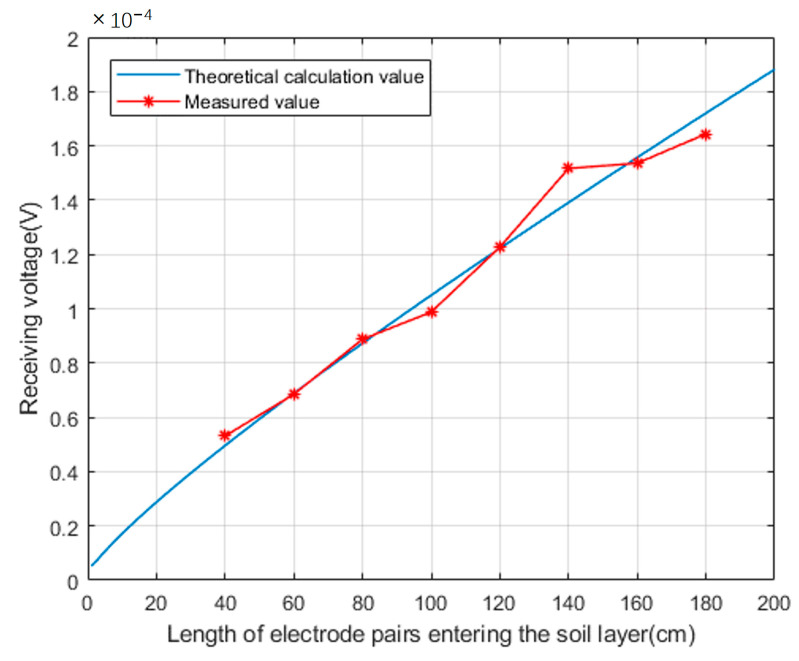
Comparison between the theoretical and measured values of the received voltage when the depth of the grounding electrode entering the soil changes.

**Figure 20 sensors-23-05936-f020:**
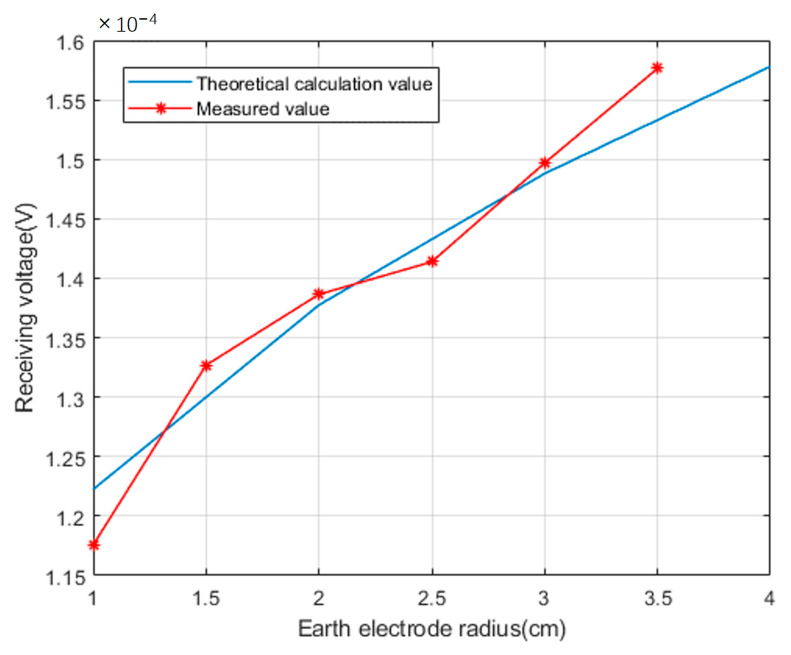
Comparison between theoretical and measured values of received voltage when the radius of the grounding electrode changes.

**Table 1 sensors-23-05936-t001:** Physical Laws of Stable Current Field and Electrostatic Field.

Electrostatic Field	Stable Current Field
Charge on each conductor in a uniform dielectric, ±Q	Current flowing through a uniform conductive medium between two electrodes, I
Electrostatic potential distribution function, V	Value Current Potential Distribution Function of Stable Current Field, V
Electric field intensity vector, E	Electric field intensity vector, E
The dielectric constant of the medium, ε	Dielectric conductivity, σ
Electric displacement vector, D=εE	Current density, j=σE

**Table 2 sensors-23-05936-t002:** Maxwell 3D modeling parameters.

Category	Parameter
Electrode radius r	0.01 m
The length of electrode pairs entering the soil layer l	1.2 m
Distance between electrodes on the same side d	4 m
Distance between opposite electrodes $L_{e}$	10 m
Number of electrode pairs n	1–4 pieces
Soil medium conductivity σ	0.01 S/m
Collapse parameters	30 m × 50 m × 15 m

**Table 3 sensors-23-05936-t003:** Relationship between the length of electrode pairs entering the soil layer and the impedance of the emitter electrode.

The Length of Electrode Pairs Entering the Soil Layer	Maxwell Simulation	Theoretical Simulation
0.6 m	99.86 Ω	100.87 Ω
1 m	56.50 Ω	57.36 Ω
1.2 m	47.42 Ω	47.66 Ω
1.5 m	37.54 Ω	38.27 Ω

**Table 4 sensors-23-05936-t004:** Relationship between the radius of the earth electrode and the impedance of the earth electrode.

Earth Electrode Radius	Maxwell Simulation	Theoretical Simulation
0.01 m	40.23 Ω	40.93 Ω
0.02 m	35.74 Ω	36.32 Ω
0.03 m	32.95 Ω	33.62 Ω
0.04 m	31.01 Ω	31.71 Ω

**Table 5 sensors-23-05936-t005:** Relationship between the number of array pairs of ground electrodes and the impedance of the ground electrode.

Number of Pairs of Ground Electrodes	Maxwell Simulation	Theoretical Simulation
1 set	40.52 Ω	40.93 Ω
2 set	20.18 Ω	20.47 Ω
3 set	13.41 Ω	13.64 Ω
4 set	10.08 Ω	10.23 Ω

## Data Availability

The dataset can be accessed upon request.

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
