# Peer review of "Research on Channel Characteristics and Electrode Electrical Performance of Earth Current Field Information Transmission Technology"

_sensors, 2023, doi:10.3390/s23135936_

Round 1
Reviewer 1 Report
1. In figure 9, 10, and 11, please provide the size of the boundaries if equivalent with the propose research.
2. At line 440, does the length and deep of the electrode improve the distance propagation if the soil is contaminated with other metal material.
3. Does the simulation include the error detection from metal in the earth soil?
4. Result in the conclusions should be highlighted in the result.
Author Response
Point 1: In figure 9, 10, and 11, please provide the size of the boundaries if equivalent with the propose research.
Response 1: When conducting finite element calculations, it is necessary to introduce a virtual boundary to truncate the infinite calculation area to a finite space. Therefore, a vacuum area is set outside the collapsed body, with the surface set as insulation, and the virtual boundary is 30m * 50m * 15m.
It has been mentioned in Table 1 of Section 5 of the manuscript.
Point 2: At line 440, does the length and deep of the electrode improve the distance propagation if the soil is contaminated with other metal material.
Response 2: According to formula , the propagation distance is only related to the resistivity of the earth's strata and the frequency of electromagnetic wave propagation. If the soil is contaminated with other metal materials and causes an increase in the resistivity of the soil in the area, the transmission distance will increase while the transmission frequency remains unchanged.
But this change is not caused by the length and depth of the electrode.
The changes in the length and depth of the electrode will cause changes in the contact impedance of the ground electrode. The decrease in impedance will increase the voltage value at the receiving end, thereby optimizing system performance.
Point 3: Does the simulation include the error detection from metal in the earth soil?
Response 3: The simulation does not include error detection of metals in soil.
Point 4: Result in the conclusions should be highlighted in the result.
Response 4: It has been modified.

Reviewer 2 Report
Issues:
1. Whether the derived results can be used into the practical system? How to utilize the derived results into practical scenario. This should be discussed in the paper.
2. Maybe the quality of the figures in this paper should be enhanced to provide more things.
3.Some papers which investigate the satellite and earth need to added into the reference section, for the satellite has a serious impact on the earth. Such as ``On the performance of the uplink satellite multi-terrestrial relay networks with hardware impairments and interference'' IEEE SJ
Author Response
Point 1: Whether the derived results can be used into the practical system? How to utilize the derived results into practical scenario. This should be discussed in the paper.
Response 1: The results obtained from this study have been applied to practical systems.
Our newly developed engineering prototype uses a transmission frequency of 10Hz, a maximum input voltage of 250V, a depth of 1.2 meters for the ground electrode, and a spacing of 4 meters. We have achieved a maximum transmission distance of 1000 meters.
In the article, practical scenario applications were added and data comparative analysis was conducted.
The following photos show the equipment we have developed and its application testing on the engineering site.

Engineering prototype photos

Off site test photos

Test photos of actual engineering scenarios
Point 2: Maybe the quality of the figures in this paper should be enhanced to provide more things.
Response 2: All formulas, charts, and icon data in this article have been modified and updated.
Point 3: Some papers which investigate the satellite and earth need to added into the reference section, for the satellite has a serious impact on the earth. Such as ``On the performance of the uplink satellite multi-terrestrial relay networks with hardware impairments and interference'' IEEE SJ
Response 3: Our research focuses on utilizing the Earth's medium to propagate electromagnetic waves, and satellites have no impact on ground penetrating communication based on the Earth's current field.

Reviewer 3 Report
In the reviewed manuscript the research on the information transmission requirements in complex geological and electromagnetic environments were presented. The manuscript is very interesting, but I have a few remarks:
In the paper the Authors have written: “Magnetic permeability is used to characterize the magnetic characteristics of a medium, as both soil and rock layers are non-magnetic media. Therefore, the earth's magnetic permeability is set to be the same as in the air, μ=4π×10-7 H/m.” In my opinion, this is too much of a simplification. Rocks may contain iron ore, and then the magnetic permeability will not be equal to the magnetic permeability of a vacuum.
The parameters of the earth medium, such as: magnetic permeability μ(H/m), dielectric constant ε(F/m) and conductivity σ(S/m) are non-linear. Therefore, the issues discussed in this article require extreme caution.
A major imperfection contained in the manuscript are incorrect formulas (15) and (17). Such imperfections should not be present in papers published in the Sensors journal. This means that all formulas contained in the paper should be checked in detail.
The most important remark is that the paper is purely computational, without measurements.
Author Response
In the reviewed manuscript the research on the information transmission requirements in complex geological and electromagnetic environments were presented. The manuscript is very interesting, but I have a few remarks:
Point 1:In the paper the Authors have written: “Magnetic permeability is used to characterize the magnetic characteristics of a medium, as both soil and rock layers are non-magnetic media. Therefore, the earth's magnetic permeability is set to be the same as in the air, μ=4π×10-7 H/m.” In my opinion, this is too much of a simplification. Rocks may contain iron ore, and then the magnetic permeability will not be equal to the magnetic permeability of a vacuum.
Response 1: The research model in this paper is established for most cases. Except for a few ferromagnetic minerals, the permeability of other minerals is very close to that of Vacuum permeability.
The model in this article is based on specific scenarios of surrounding rock masses and collapses in tunnel engineering, and there is no iron ore in the tunnel engineering crossing zone in the general transportation field; If iron ore is present in rocks in other scenarios and there is a significant difference in magnetic permeability, it is necessary to change the magnetic permeability value.
Point 2:The parameters of the earth medium, such as: magnetic permeability μ(H/m), dielectric constant ε(F/m) and conductivity σ(S/m) are non-linear. Therefore, the issues discussed in this article require extreme caution.
Response 2: The earth's medium is complex and variable. The basis for establishing the model in this article is to assume that the earth is a homogeneous medium.
Point 3:A major imperfection contained in the manuscript are incorrect formulas (15) and (17). Such imperfections should not be present in papers published in the Sensors journal. This means that all formulas contained in the paper should be checked in detail.
Response 3: All formulas, charts, and icon data have been checked and confirmed to be correct.
Point 4:The most important remark is that the paper is purely computational, without measurements.
Response 4: Actual measurement data for on-site scenario application has been added.